# Leveraging Low-Rank Structure for Effective Weight-Sharing in Language Models

**Mark Muchane**[1]**, George Sokolik**[2]**, Micah Goldblum**[3,*]**Sanae Lotfi**[4,*]
[1]University of Chicago
[2]Imperial College London
[3]Columbia University
[4]Meta Superintelligence Labs

## Abstract

Small language models are typically built by heuristically scaling down the architectures of large language models. We investigate whether small models can be parameterized more effectively by sharing weights across attention heads and transformer layers while capturing their differences with low rank adaptation modules. To understand the limits and tradeoffs of this approach, we conduct controlled pretraining experiments that compare several weight sharing strategies under strict parameter-matched constraints across four model scales from 100M to 1B parameters. We find that attention matrices, and even entire transformer layers, can be shared without degrading performance, though overly aggressive sharing configurations yield diminishing or negative returns. Within the effective sharing regime, weight sharing deliberately trades increased FLOPs per parameter for a reduced memory footprint, matching or improving over parameter-matched unshared baselines. We also explore reducing the parameter cost of the embedding layer through a factorized construction, which yields additional memory savings and enables more effective parameter allocation. To motivate these design choices, we analyze the effective rank of model weights and the residual stream. Our analysis, along with downstream evaluations, provides a recipe for designing more efficient compact models.

## 1 Introduction

Large language models (LLMs) have made consistent gains across a range of upstream and downstream tasks by scaling the number of parameters across model iterations (Kaplan et al., 2020; Brown et al., 2020). However, extremely large models incur substantial training and inference costs, including compute, memory, and energy requirements. In response, there is growing interest in strong small ($\leq$ 1B parameter) language models (Allal et al., 2024; Team, 2025; Yang et al., 2025). These models have a smaller memory footprint, are more suitable for local and on-device deployment, and are more accessible in settings with constrained hardware.

Small language models are often obtained by scaling down standard large language model architectures, e.g., shrinking hidden dimensions, number of layers, or attention heads. In this work, we ask whether such scaled-down designs miss a simple opportunity: many weight tensors may be similar enough that they can be shared, with only low-rank differences needed to recover model capacity. Our central hypothesis is that a substantial fraction of weight differences in pretrained language models are well-approximated as low-rank updates. If so, we can share base weights and represent head- or layer-specific variation using LoRA modules (Hu et al., 2022).

The goal of this work is to explore lightweight LLM architectural variants that optimize performance per parameter. This goal differs from FLOPs-efficiency. Our methods will increase FLOPs per parameter, whereas FLOPs-efficient methods like Mixture of Experts may decrease FLOPs per parameter. In memory-constrained settings, paying extra FLOPs may be acceptable as long as the parameter count stays small.

---

*Equal last authorship contribution.

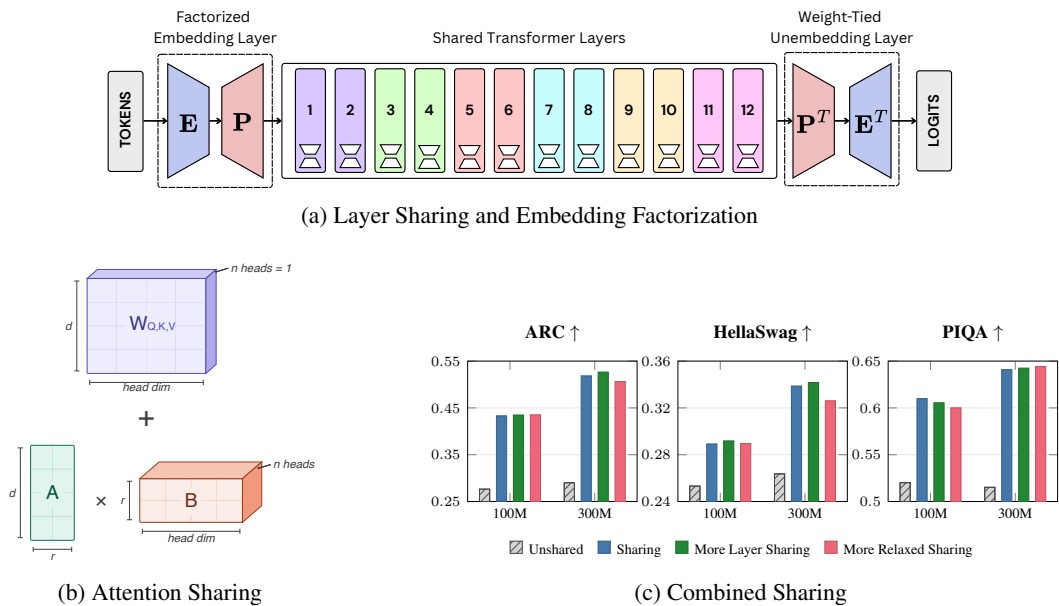

Figure 1: **Low rank structure can be used to effective weight-shared LLMs (a)** Embeddings and embeddings can be tied and factorized, layers can be shared with a low-rank difference (weight-sharing indicated by shared color); **(b)** Attention projections can be re-parameterized with heads only differing by a low-rank offset; **(c)** Combining these techniques yields more performant small LLMs. At 100M and 300M parameters, combined sharing models beat unshared on a variety of downstream benchmarks.

We take a two-step approach where we first analyze a pretrained model, Gemma 3-270M (Team, 2025), and measure the effective rank of weight and activation differences using singular value decomposition (SVD). This analysis suggests that attention head weights, and differences between some early layer outputs, have low effective rank. Second, guided by these observations, we run controlled pretraining experiments that compare different sharing strategies for attention and for transformer layers. To keep comparisons fair, we evaluate each sharing strategy against parameter-matched unshared baselines, carefully tuning the rank of the LoRA modules based on our SVD diagnostics.

Across these experiments, we find that weight sharing combined with low-rank adapters can reduce parameters while maintaining, and in some settings improving, loss relative to parameter-matched baselines. We also show that this approach improves performance on downstream reasoning tasks. At the same time, our experiments reveal clear limits: overly aggressive sharing configurations, such as combining head sharing with cross QKV sharing or using very deep multipliers at the 1B scale, degrade performance. Finally, we address the embedding bottleneck in sub-billion models by using factorized embeddings with weight tying (Press & Wolf, 2017; Lan et al., 2020), which reallocates parameters from the embedding matrices to other components. Both the positive results and the failure modes provide a practical recipe for improving parameter efficiency in small language models, and they connect to recent theory work that uses low-dimensional parameterizations to obtain tighter generalization bounds in large language models (Lotfi et al., 2023; 2024).

## 2 RELATED WORK

**Small language models.** The demand for efficient on device deployment has driven rapid progress in sub-billion parameter models such as Gemma (Team, 2025), Qwen (Yang et al., 2025) , and SmolLM (Allal et al., 2024). A notable recent advance is MobileLLM (Liu et al., 2024), which demonstrates that deep and narrow architectures coupled with immediate block-wise weight sharing establish strong baselines for small models. While MobileLLM replicates exact weights across adjacent blocks, our work differs by introducing low rank adapters to recover layer-specific capacity, exploring

non adjacent sharing topologies and benchmarking these strategies under strict parameter-matched constraints.

**Efficient embeddings.** The embedding and output projection layers can dominate the parameter budget of small language models, often creating a bottleneck. Weight tying (Press & Wolf, 2017) shares the input embedding and output projection matrices, and factorized embeddings (Lan et al., 2020) decompose the embedding matrix into two smaller factors to decouple vocabulary size from model dimension. Similar factorized approaches have been explored in architectures like the Sub-former (Reid et al., 2021). Recent work on vocabulary scaling laws shows that larger vocabularies are generally beneficial for performance (Tao et al., 2024; Takase et al., 2024; Huang et al., 2025), which makes reducing the per token parameter cost all the more important. We build on these techniques by sweeping the factorized embedding rank across four model sizes and identifying a more efficient, prescriptive allocation regime than the default used by most small model designs.

**Parameter sharing across depth.** Sharing weights across transformer layers was explored by Dehghani et al. (2019) who showed that a Universal Transformer in which all layers share a single set of weights can outperform standard transformers on certain tasks. Takase & Kiyono (2023) later showed that less rigid schemes such as pairwise sharing of adjacent layers often perform better than full sharing. Recent methods have proposed dynamic layer tying using reinforcement learning during training (Hay & Wolf, 2024) or cross layer parameter sharing via singular value decomposition for model compression (Wang et al., 2024). More recently Bae et al. (2024) added low rank adaptation modules (Hu et al., 2022) to shared layer models to recover some layer specific flexibility. We go further by using a low rank analysis of pretrained weights to choose which components to share, by introducing two new sharing topologies (Hierarchy and Bulge) motivated by the observation that boundary layers play distinct roles (Gromov et al., 2024), and by testing all strategies under strict parameter matched pretraining comparisons.

**Low rank structure in transformers.** Several works have observed that attention outputs and weight matrices in trained transformers exhibit low rank structure (Wang et al., 2025). Low rank adaptation (Hu et al., 2022) exploits this observation for efficient finetuning by freezing pretrained weights and adding trainable low rank updates. Recent work extends this to pretraining and inference, such as using matrix based dictionary learning for attention weights (Zhussip et al., 2026), cross layer singular value decomposition for key value cache compression (Chang et al., 2025), and exploring the trade-offs of combining parameter-efficient experts fine-tuned using low rank adaptation (Lotfi et al., 2026). Similarly, O'Neill et al. (2026) introduce Low Rank Key Value attention to reduce memory footprint, and DeltaLLM compresses language models after training by sharing weights across layers and modeling the differences with low rank deltas (Mikaelyan et al., 2025). Other work has shown that while low-dimensional projections in standard attention can cause information loss, structured matrices can recover this capacity (Kuang et al., 2025).

Our work differs in that we apply low rank adapters to represent differences between shared base weights during pretraining from scratch rather than for compression after training, and we evaluate sharing across both attention heads and transformer layers.

**Scaling laws and model design.** Scaling laws (Kaplan et al., 2020) and compute optimal training regimes (Hoffmann et al., 2022) provide guidance on how to allocate compute across model size and data. When optimizing architectures under fixed parameter budgets, recent studies show that deep and narrow models often generalize better and learn more efficiently than wider, shallower counterparts (Tay et al., 2021; Petty et al., 2024). However these findings are typically derived for standard architectures and do not account for weight sharing. Our experiments contrast with standard scaling assumptions by demonstrating that weight sharing shifts the optimal design point, allowing models to be significantly deeper without increasing the parameter count, which provides a principled recipe for settings where memory footprint rather than compute is the binding constraint.

## 3 LOW-RANK STRUCTURE OF LLMS

Before designing sharing strategies, we analyze the internal structure of a pretrained small language model to identify which components exhibit low rank structure and are therefore amenable to sharing. We use Gemma 3 270M (Team, 2025) as our reference model and conduct two sets of analyses. In Section 3.1, we decompose weight and activation differences across attention heads and layers to

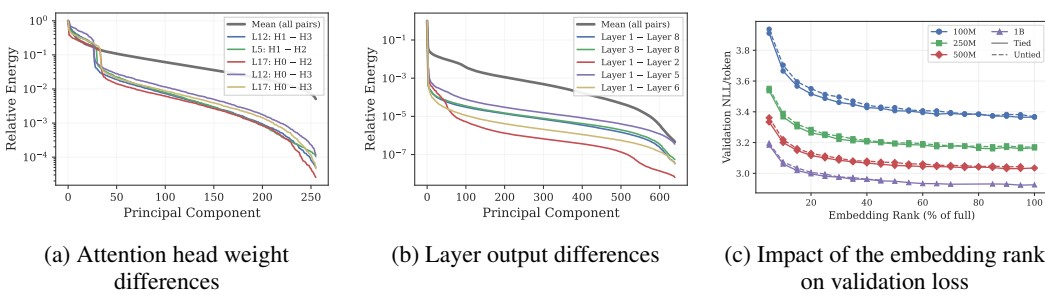

(a) Attention head weight differences

(b) Layer output differences

(c) Impact of the embedding rank on validation loss

Figure 2: **Major parts of LLMs exhibit low rank structure. (a)** Attention head weight differences require few principal components to capture 95% of variance. **(b)** Early non-adjacent layer output differences are similarly low-rank. **(c)** The embedding matrix can be reduced to half-rank before loss goes up significantly. These results inform the use of shared base weights with LoRA and factorized embeddings to save expressivity while preserving a fixed parameter budget.

measure where low rank structure is most pronounced. In Section 3.2, we test whether the embedding matrix itself is low rank by progressively factorizing it and measuring when performance degrades, and whether tying the input and output embeddings preserves or harms quality at each rank. These analyses motivate the sharing and factorization strategies we evaluate in Section 4.

### 3.1 LOW RANK STRUCTURE IN WEIGHTS AND ACTIVATIONS

We begin by analyzing which components of a pretrained small language model have the most similar weight tensors, since these are the best candidates for sharing. We use Gemma 3 270M (Team, 2025) as our reference model and measure pairwise differences using singular value decomposition (SVD). Given a matrix $\widetilde{A} \in \mathbb{R}^{n \times d}$, its SVD is $\widetilde{A} = U \Sigma V^\top$ where $U \in \mathbb{R}^{n \times r}$, $V \in \mathbb{R}^{d \times r}$, and $\Sigma = \mathrm{diag}(\sigma_1, \ldots, \sigma_r)$ contains the singular values in decreasing order. The squared singular values $\sigma_i^2$ indicate the variance captured along each principal direction, and we use the number of components required to capture 95% of the total variance as our measure of effective rank.

**Attention head differences.** We compute pairwise differences between all attention weight matrices within each layer and decompose each difference matrix. As shown in Figure 2a, the five pairs with the lowest effective rank are all differences between attention head weights rather than differences between the query, key, and value projections. This is consistent with the observation that Gemma 3 270M uses multiple heads only for the query matrix, with key and value each having a single head of dimension 256. The low rank structure of these head differences suggests that attention heads can share a common base weight with only a small rank correction needed to recover head specific behavior. The number of components required to explain 95% of the variance in these differences also provides a principled guide for choosing the adapter rank in our attention sharing experiments in Section 4.2.

**Layer output differences.** For cross layer analysis, we take the difference between layer outputs, that is, what each layer adds to the residual stream, computed over 25,000 tokens of pretraining data to capture the full effect of each layer rather than a single transformation. As shown in Figure 2b, we find that non adjacent early layers such as layers 1 and 5 have the lowest rank differences. This finding is notably different from that of Gromov et al. (2024) and suggests that more aggressive sharing in later layers may not be effective. Because four out of five of the most similar layer pairs are non adjacent, this also suggests that a sharing strategy pairing non adjacent layers may outperform the sequential pairing used in prior work. This observation directly informs the design of our Bulge sharing topology in Section 4.3, and the effective rank of these layer differences guides our choice of adapter rank in the layerwise sharing experiments.

### 3.2 LOW RANK STRUCTURE IN THE EMBEDDING MATRIX

Given that the weight matrices and residual stream exhibit low rank structure, a natural question is whether the embedding matrix does as well. This question is particularly relevant for small language models, where embedding related parameters can constitute a disproportionately large fraction of the

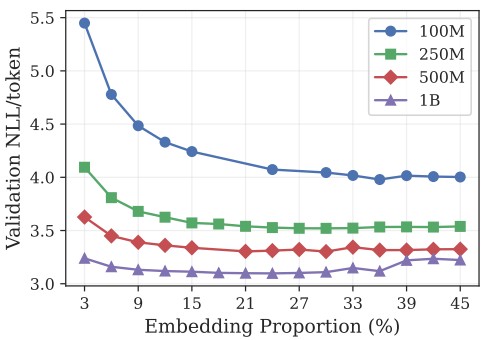 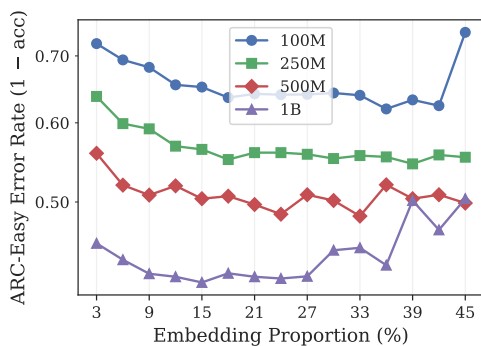

(a) Val NLL/token as a function of embedding proportion.

(b) ARC-Easy error rate as a function of embedding proportion across model sizes.

Figure 3: **Optimal embedding proportion on val. loss and downstream tasks.** All configurations are parameter-matched. Despite high embedding proportions appearing to be effective in reducing loss, they have little effect on downstream model performance.

total parameter count. For instance, the embedding table in Gemma 3 270M accounts for roughly 170M of the model's 270M parameters, or over 63% of the total (Team, 2025).

If the embedding matrix is effectively low rank, factorizing it would free a large number of parameters that could be reallocated to the transformer body. To test this, we factorize the embedding matrix from 5% to 100% of its full rank in increments of 5% and measure validation loss at each point. As shown in Figure 2c, loss does not increase meaningfully until the rank drops below roughly 50% of the full embedding dimension, confirming that the embedding matrix is substantially overparameterized in standard small models.

We also compare tied and untied factorized embeddings at each rank. If weight tying was harmful, we would expect an untied embedding at rank $r$ to perform comparably to a tied embedding at rank $2r$, since tying effectively halves the number of free embedding parameters. In practice, untied embeddings perform worse than tied embeddings at the same rank, indicating that tying acts as a beneficial constraint rather than a capacity bottleneck. These observations motivate our use of tied factorized embeddings in Section 4.1.

## 4 WEIGHT SHARING TRADEOFFS IN SMALL LANGUAGE MODELS

We now evaluate the sharing and factorization strategies motivated by the analysis in Section 3.1 and Section 3.2. All models follow a Llama 3 architecture (Dubey et al., 2024; Meta AI, 2024) and are trained from scratch using Meta's Lingua package (Videau et al., 2024) on 20 billion tokens (representing a mix of web, math, and coding data), which is at least Chinchilla optimal for the 1B scale (Hoffmann et al., 2022). We train at four parameter scales: 100M, 250M, 500M, and 1B. Each sharing configuration is compared against a parameter matched unshared baseline that removes the same number of parameters by uniformly shrinking model dimensions, so that any improvement can be attributed to the sharing strategy rather than to a difference in total capacity. We evaluate on validation loss (negative log likelihood per token) and on three downstream reasoning benchmarks: ARC Easy (Clark et al., 2018), HellaSwag (Zellers et al., 2019), and PIQA (Bisk et al., 2020).

### 4.1 FACTORIZED EMBEDDINGS

For models at the sub-billion parameter scale, embedding related parameters can constitute a disproportionately large fraction of the total parameter count. While using a smaller vocabulary could reduce this cost, larger vocabularies provide significant representational and compute advantages by encoding more information into a single token (Tao et al., 2024; Takase et al., 2024; Huang et al., 2025). We therefore keep the vocabulary fixed and instead reduce the per token parameter cost through factorization and weight tying.

We decompose the embedding matrix $W_{emb}$ of size $V \times d$ into two smaller matrices: a low rank embedding matrix $E$ of size $V \times r_e$ and a projection matrix $P$ of size $r_e \times d$, where $r_e \ll d$ (Lan et al., 2020). A token is first mapped to a low dimensional space by $E$ and then projected up to the model dimension by $P$. This reduces the embedding parameter cost $N_{emb}$ from $V \times d$ to $r_e(V + d)$, and the rank $r_e$ provides fine grained control over how many parameters are allocated to the embedding versus the transformer body. We further reduce embedding related parameters by tying the input embedding and output projection matrices (Press & Wolf, 2017), so that the output projection becomes $(EP)^\top$. As confirmed in Section 3.2, tying acts as a beneficial constraint rather than a capacity bottleneck.

Unlike the sharing experiments in the following subsections, the factorized embedding sweep holds the total parameter count $N_{tot}$ fixed for each model size, compensating for any reduction in embedding parameters by widening the feed forward layers or adding depth. This isolates the effect of parameter allocation between the embedding and the transformer body. We define the embedding proportion $\rho = N_{emb}/N_{tot}$ and sweep it across 100M, 250M, 500M, and 1B with tied factorized embeddings. When evaluated against validation loss in Figure 3a, we observe a clear optimum around 21 to 27% depending on scale, with loss monotonically improving at 100M and 250M and a distinct minimum appearing at 500M and 1B beyond which the narrower transformer degrades performance.

Downstream evaluation on ARC Easy in Figure 3b tells a somewhat different story: performance degrades sharply when $\rho$ falls below roughly 12%, improves as $\rho$ increases to 15 to 24%, and then plateaus or reverses at higher proportions, particularly at 1B. The gap between the validation loss optimum and the downstream optimum suggests that allocating too many parameters to embeddings can improve perplexity without translating to reasoning gains. In either case, the default embedding proportions used by most small model designs are substantially higher than optimal, and reallocating parameters from the embeddings to the transformer body improves both metrics.

## 4.2 ATTENTION SHARING

Motivated by the low rank structure of attention head differences observed in Section 3.1, we explore sharing a single base weight matrix across all attention heads within each layer, with per head low rank adapters to recover head specific behavior. Concretely, we define a shared base weight $W_{\text{base}}$ and express each head's weight as:

$$W_i = W_{\text{base}} + A_i B_i \tag{1}$$

where $A_i \in \mathbb{R}^{d \times r}$ and $B_i \in \mathbb{R}^{r \times d_{head}}$ are the low rank adapter matrices for head $i$, and $r$ is the adapter rank. This formulation generalizes Group Query Attention (Ainslie et al., 2023; Shazeer, 2019): when $r = 0$ all heads are identical, and as $r$ grows toward $d_{head}$ the heads recover full independence. The adapter rank therefore provides a continuous dial between strict sharing and no sharing at all, and the SVD analysis in Section 3.1 suggests that a modest rank suffices to capture the variance between heads.

**Head sharing.** As shown in Figure 4a, sharing attention heads with low rank adapters matches models that are millions of parameters larger and consistently beats the parameter matched unshared baseline across scales. We can further tighten the sharing adjusting rank–and we conduct a full sweep of head sharing at relatively high and low-ranks across model sizes in Table 1.

**Rank sensitivity.** We sweep the adapter rank across a wide range of values at each model scale without holding total parameter count consistent, to evaluate if there are diminishing returns to higher rank sharing. Validation loss improves monotonically with rank but with clear diminishing returns (Figure 6), consistent with the SVD analysis showing that most of the variance between heads is concentrated in the first few singular values. Downstream accuracy on ARC Easy, however, does not change significantly across all ranks and scales as shown in Figure 7, indicating that the additional capacity purchased by higher ranks translates to perplexity gains but not to measurable reasoning improvements. We observe this most clearly in the parameter matched setting where allocating additional parameters to the adapter matrices rather than the main attention projections yields strong validation loss but costs one downstream evaluation point. This disconnect between validation loss and downstream performance echoes the pattern observed in the embedding proportion sweep (Section 4.1) and suggests that practitioners can use a relatively low rank without sacrificing downstream quality.

**QKV sharing.** We also test a more aggressive form of sharing where a single base weight is shared across all of the query, key, and value projection matrices, with each differing from the base by

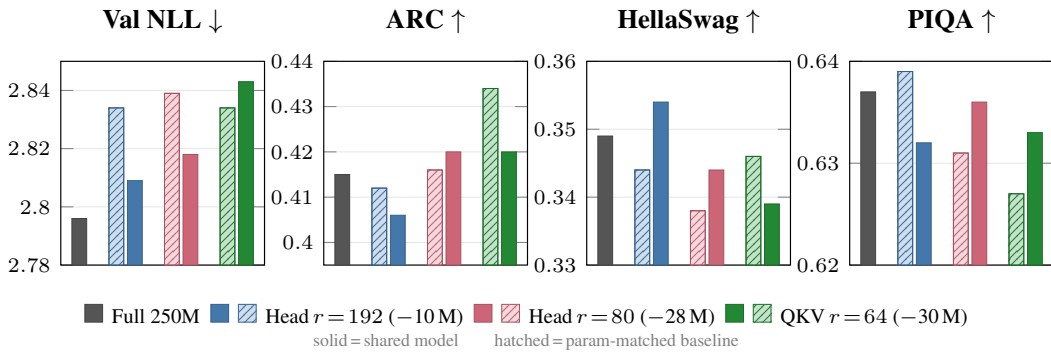

(a) **Attention sharing.** Each color pairs a shared model (solid) with its parameter-matched baseline (hatched). Head sharing at $r = 80$ ($-28$ M) consistently meets or exceeds its baseline, nearly matching the full 250M model. QKV sharing ($-30$ M) shows mixed downstream results.

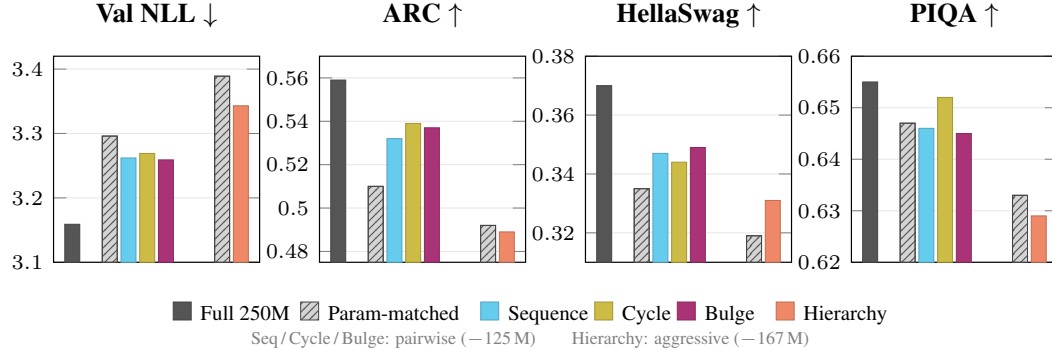

(b) **Layer sharing.** Gray hatched bars are parameter-matched baselines. All three pairwise strategies ($-125$ M) outperform their matched baseline on downstream tasks despite large parameter reductions. Hierarchy ($-167$ M) remains competitive with its own matched baseline.

Figure 4: **Weight sharing results at 250M scale.** Solid bars denote shared models; hatched bars denote parameter-matched unshared baselines.

a low rank adapter in addition to sharing the heads. Sharing all of Q, K, and V does reduce the overall parameter count and delivers a loss increase that is smaller than naively removing the same number of parameters, but this configuration may be too aggressive. Downstream evaluations are more optimistic than validation loss here, suggesting that the loss degradation overstates the practical impact on reasoning tasks.

### 4.3 LAYERWISE SHARING

Weight sharing across transformer layers presents a compelling mechanism for reducing the parameter footprint of a model. Strict layer sharing, however, imposes representational bottlenecks and limits a model's capacity to learn layer specific transformations. This constraint can be relaxed with the addition of low rank adapter modules into the shared attention and feed forward sublayer weight matrices, allowing for layer specific specialization while maintaining a reduced parameter count.

We start by entirely decoupling the low rank relaxation to investigate sharing strategy in isolation. Determining which sets of layers to share within the model is a central design choice. Figure 5 details the key strategies investigated. The standard transformer configuration, **Vanilla** (Figure 5a), uses $L$ layers. Takase & Kiyono (2023) proposed **Sequence**, **Cycle**, and **Cycle (Rev)** (Figures 5b to 5d) with the number of unique layers for each strategy effectively becoming $L/2$. These strategies outperformed both Vanilla and Universal configurations on machine translation while exhibiting reduced training times.

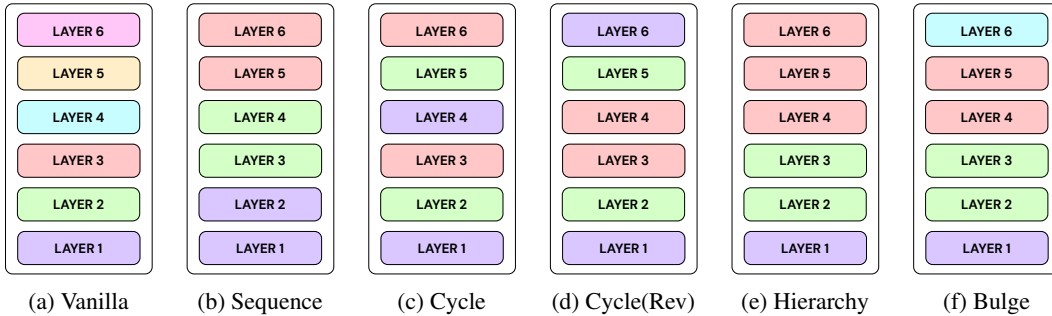

(a) Vanilla    (b) Sequence    (c) Cycle    (d) Cycle(Rev)    (e) Hierarchy    (f) Bulge

Figure 5: **Comparison of different layer-sharing architectural strategies.** Shared layer weights are denoted by shared colour. We show 6-layer models for illustrative purposes.

Prior work suggests that different depths of transformer layers serve distinct representational roles. While Gromov et al. (2024) highlight the contribution of earlier layers to overall performance, Gardinazzi et al. (2024) demonstrate that boundary layers, both initial and final, exhibit unique functional characteristics compared to intermediate layers. Our SVD analysis in Section 3.1 supports this observation: the layer output differences with the lowest effective rank are concentrated among non adjacent early layers, while later and boundary layers are harder to approximate with a low rank strcture. Motivated by these findings, we introduce **Hierarchy** (Figure 5e) and **Bulge** (Figure 5f). Hierarchy uses asymmetric sharing, preserving unique parameters in the earliest layers where our SVD analysis shows the most redundancy. Bulge isolates both the first and final boundary layers, concentrating the sharing within the intermediate layers. where the low rank structure is most pronounced.

**Sharing strategy results.** Our results for strict layer sharing (no adapters) show that no singular sharing pattern dominates on every metric and scale. Across model sizes, Cycle is the strongest overall strategy on downstream benchmarks, whereas Sequence typically gives the best validation loss (or is tied for best at larger scales, shown in Appendix B, Table 2). Bulge is competitive but does not consistently outperform Cycle or Sequence, while Hierachy is generally less effective with its more aggressive sharing. These results indicate that the choice of sharing pattern does indeed matter, but do not support a blanket conclusion that boundary layers should remain unshared.

**Rank sensitivity.** Here, we reintroduce low rank adapters and sweep the adapter rank (using Bulge as the chosen topology to do so). As with attention sharing, validation loss improves monotonically with rank but with diminishing returns as in Figure 8. Unlike attention sharing, downstream performance does show meaningful improvement with rank at the larger scales as shown in Figure 9, particularly at 500M and 1B. This suggests that layer sharing operates in a higher capacity regime than head sharing, where the adapter rank has a more direct effect on the model's ability to solve reasoning tasks. The gap between validation loss and downstream accuracy is narrower here, indicating that the parameters invested in layer specific adapters translate more efficiently to downstream quality.

**Scaling depth with sharing.** Layer sharing also opens the possibility of building deeper models at a fixed parameter budget. We test hypothesis this by increasing the total number of layers while keeping the number of unique layer weights fixed, effectively using the freed parameters to add depth rather than width, shown in Table 2 and Table 3. These deeper shared models outperform their shallower unshared counterparts at the same parameter count, demonstrating that layer sharing provides a practical mechanism for trading width for depth. The benefit is most pronounced at smaller scales where the baseline models have relatively few layers and the additional depth provides meaningful representational gains.

## 5    COMBINING SHARING STRATEGIES

Having established that sharing attention heads, sharing transformer layers, and factorizing embeddings each can yield parameter efficiency gains in isolation, we now test whether these strategies can be combined. If the low rank structure we observed in Section 4 holds across multiple dimensions

simultaneously, a maximally shared model should strictly outperform an unshared baseline at the same parameter count by reallocating its budget toward deeper or wider architectures.

We train 4 configurations at the 100M and 300M parameter scales. To keep comparisons fair, all models are strictly parameter matched and trained from scratch on 20 billion tokens. We have a unshared baseline that represent standard architectural choices (matching the 600M model hyperparameters at uniform scaling). The remaining three are shared configurations that combine factorized embeddings, attention sharing, and various layerwise sharing strategies. We evaluate on three downstream reasoning benchmarks: ARC Easy, HellaSwag, and PIQA, reporting accuracy in Figure 1c.

**Composing sharing strategies.** Across both scales, every shared configuration outperforms every unshared baseline on average. At 100M, the best shared model (More Layer Sharing) exceeds the unshared baseline by +9.4% on average. This improvement is consistent across tasks, with gains of +15.9 on ARC-Easy, +8.5 on PIQA, and +3.9 on HellaSwag. The strong performance of a model that uses sharing to increase depth is confirmed by these results.

**Scaling and robustness** The performance gap between shared and unshared models widens as we scale from 100M to 300M parameters. This suggests that as small models become less constrained overall, sharing is better able extract more effective capacity per unique parameter. We also find that the exact choice of sharing topology is less important than the decision to share weights in the first place. At 300M, even the worst performing shared configuration (standard sharing) still exceeds the best unshared baseline by +12.7%.

## 6 CONCLUSION

In this work, we investigated the extent to which weight sharing reduces the parameter cost of small language models without degrading their quality. Our controlled experiments across four scales from 100M to 1B show that sharing attention heads and entire transformer layers through low rank adapters matches or outperforms parameter matched unshared baselines on validation loss, while factorized embeddings with weight tying recover most of the parameters otherwise consumed by the embedding matrix. The tradeoffs, however, are not symmetric across sharing types.

When all three strategies are combined at the 100M and 300M scales, shared configuration outperforms unshared baseline, and the gap widens with scale. These results confirm that moderate, targeted sharing increases effective capacity per unique parameter and provide a practical recipe for designing compact models under strict memory constraints.

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

# A    EXTENDED RESULTS

All models follow the Llama 3 architecture (Dubey et al., 2024) and are trained with Meta's Lingua framework (Videau et al., 2024). We use the AdamW optimizer with a learning rate of $3 \times 10^{-4}$, selected by sweeping one order of magnitude above and below and choosing the setting with the lowest validation loss. Training uses a cosine schedule with 2 000 warmup steps, gradient clipping at 10.0, bf16 mixed precision, and a context length of 2,048 tokens. All runs use 8 GPUs without model sharding. All models use GQA and SwiGLU activations with RMSNorm.

Table 1 shows the extended validation and downstream evaluation results for attention sharing across all model scales, extending the results from the 250M scale seen in the main body in Figure 4a. Results for both head sharing and QKV sharing at different LoRA ranks are given in Table 1.

Table 2 shows the extended validation and downstream evaluation results for layer sharing across all model scales, extending the results from the 250M scale seen in the main body in Figure 4b. Note that experimentation with model aspect ratio's (Deep vs Wide vs Standard) was initially only done at the 100M scale and then extended towards a full suite of subsequent training runs across model sizes (seen in Table 3). It should also be noted that the reported base model sizes (100M, 250M, 500M, and 1B) refer to the number of non-embedding parameters, rather than total parameter count, given that this experiment is only concerned with the transformer layers of a model, and thus only concerns transformer-based parameters.

Table 3 shows experimentation with model aspect ratios to observe the impact of model depth when layer sharing is enabled. The initial experimentation done at 100M in Table 2 is now extended to include all model scales, additionally, model depth is also extended to show performance from a $1\times$ multiplier in unshared models to $\geq 3\times$ for shared models at maximum depth.

# B    LoRA RANK SWEEPS

In order to isolate the effect of adapter rank from other attention sharing design choices and model hyperparameters, we fix the model to employ the Head sharing variant at all model scales and sweep LoRA rank while parameter-matching between individual LoRA ranks for each model scale. Figure 6 and Figure 7 show the resultant performance from the LoRA rank sweep via validation loss and downstream evaluation on ARC-Easy respectively.

Similarly, in Figure 8 and Figure 9, across all investigated model scales, we fix the respective models to utilise Bulge style sharing and sweep the LoRA rank across each shared layer to show validation loss (Figure 8) and downstream evaluation on ARC-Easy(Figure 9).

Table 1: **Attention sharing with parameter-matched baselines across all model sizes.** Shared models use low-rank attention weight sharing. "Head Shared" shares attention heads with a LoRA adapter of rank $r$; "QKV Shared" shares all of Q, K, V projections. $\Delta$Params is relative to the full-size unshared baseline in each group. Delta columns compare each shared model to its parameter-matched unshared baseline. **Bold** marks the best result in each size group.

| Configuration | $\Delta$Params | Val NLL | ARC | Hella | PIQA | $\Delta$ARC | $\Delta$Hella | $\Delta$PIQA |
|---|---|---|---|---|---|---|---|---|
| *100M scale* | | | | | | | | |
| Unshared, 92M | — | 3.006 | .397 | .307 | .604 | — | — | — |
| Unshared, 85M | $-7$M | 3.020 | **.402** | .301 | .604 | — | — | — |
| Unshared, 83M | $-9$M | 2.998 | .383 | .308 | .606 | — | — | — |
| Head Shared, $r=144$ | 0 | **2.972** | .394 | **.315** | .607 | $-.003$ | $+.008$ | $+.003$ |
| Head Shared, $r=22$ | $-7$M | 2.998 | .395 | .308 | .614 | $-.006$ | $+.007$ | $+.010$ |
| QKV Shared, $r=4$ | $-9$M | 3.056 | .370 | .296 | **.621** | $-.013$ | $-.012$ | $+.015$ |
| *250M scale* | | | | | | | | |
| Unshared, 250M (full) | — | **2.796** | .415 | .349 | **.637** | — | — | — |
| Unshared, 240M | $-10$M | 2.834 | .412 | .344 | .639 | — | — | — |
| Unshared, 222M | $-28$M | 2.839 | .416 | .338 | .631 | — | — | — |
| Unshared, 220M | $-30$M | 2.834 | **.434** | .346 | .627 | — | — | — |
| Head Shared, $r=192$ | $-10$M | 2.809 | .406 | **.354** | .632 | — | — | — |
| Head Shared, $r=80$ | $-28$M | 2.818 | .420 | .344 | .636 | $+.005$ | $+.006$ | $+.005$ |
| QKV Shared, $r=64$ | $-30$M | 2.843 | .420 | .339 | .633 | $-.013$ | $-.006$ | $+.006$ |
| *500M scale* | | | | | | | | |
| Unshared, 500M (full) | — | **2.688** | .445 | **.395** | **.653** | — | — | — |
| Unshared, 460M | $-40$M | 2.724 | **.452** | .381 | .647 | — | — | — |
| Unshared, 440M | $-60$M | 2.731 | .439 | .378 | .652 | — | — | — |
| Unshared, 435M | $-65$M | 2.717 | .443 | .380 | .645 | — | — | — |
| Head Shared, $r=192$ | $-40$M | 2.706 | .446 | .386 | .652 | $-.006$ | $+.005$ | $+.005$ |
| Head Shared, $r=80$ | $-60$M | 2.724 | .458 | .383 | .658 | $+.019$ | $+.005$ | $+.006$ |
| QKV Shared, $r=64$ | $-65$M | 2.743 | .432 | .376 | .652 | $-.011$ | $-.004$ | $+.007$ |
| *1B scale* | | | | | | | | |
| Unshared, 1B (full) | — | **2.651** | .460 | **.427** | .669 | — | — | — |
| Unshared, 945M | $-55$M | 2.672 | .460 | .422 | .667 | — | — | — |
| Unshared, 919M | $-81$M | 2.681 | .449 | .418 | .662 | — | — | — |
| Unshared, 907M | $-93$M | 2.668 | .463 | .415 | .665 | — | — | — |
| Head Shared, $r=192$ | $-55$M | 2.662 | **.464** | .419 | **.672** | $+.004$ | $-.002$ | $+.005$ |
| Head Shared, $r=80$ | $-81$M | 2.669 | .446 | .424 | .663 | $-.004$ | $+.006$ | $+.002$ |
| QKV Shared, $r=64$ | $-93$M | 2.696 | .425 | .405 | .655 | $-.039$ | $-.010$ | $-.010$ |

Table 2: **Layer sharing strategy comparison across model sizes.** All shared models double the depth (using half the unique layers with sharing) at the same width as the unshared baseline. Parameter-matched unshared baselines are trained at reduced width to match the effective parameter count of shared models. At 100M, three architectural variants are tested: (Standard)—same width as baseline, (Wide)—wider but shallower, and (Deep)—narrower but deeper. Delta columns compare each shared model to its corresponding parameter-matched unshared baseline. **Bold** marks the best result in each size group.

| Strategy | Variant | Val NLL | ARC | Hella | PIQA | ΔARC | ΔHella | ΔPIQA |
|---|---|---|---|---|---|---|---|---|
| *100M scale — full-size unshared baseline* | | | | | | | | |
| Unshared | (Full, 100M) | **3.363** | **.505** | **.324** | .626 | — | — | — |
| *100M scale — param-matched baselines (≈50M unique params)* | | | | | | | | |
| Unshared, 50M | (Standard) | 3.530 | .464 | .297 | .614 | — | — | — |
| Unshared, 50M | (Wide) | 3.532 | .461 | .297 | .620 | — | — | — |
| Unshared, 50M | (Deep) | 3.516 | .465 | .299 | .615 | — | — | — |
| Sequence | (Standard) | 3.478 | .473 | .307 | .616 | +.010 | +.009 | +.002 |
| Sequence | (Wide) | 3.550 | .460 | .297 | .617 | −.001 | +.000 | −.003 |
| Sequence | (Deep) | 3.432 | .480 | .308 | .629 | +.016 | +.009 | +.014 |
| Cycle | (Standard) | 3.495 | .462 | .301 | .623 | −.002 | +.004 | +.009 |
| Cycle | (Wide) | 3.555 | .471 | .298 | .619 | +.010 | +.001 | −.002 |
| Cycle | (Deep) | 3.444 | .493 | .308 | **.633** | +.028 | +.009 | +.018 |
| Cycle-Rev | (Standard) | 3.491 | .490 | .302 | .621 | +.026 | +.004 | +.007 |
| Cycle-Rev | (Wide) | 3.561 | .463 | .296 | .614 | +.002 | −.001 | −.006 |
| Cycle-Rev | (Deep) | 3.443 | .485 | .308 | .632 | +.021 | +.009 | +.017 |
| Bulge | (Standard) | 3.484 | .471 | .304 | .626 | +.008 | +.006 | +.011 |
| Bulge | (Wide) | 3.500 | .467 | .309 | .630 | +.006 | +.012 | +.010 |
| Bulge | (Deep) | 3.486 | .484 | .305 | .624 | +.019 | +.005 | +.008 |
| *100M scale — Hierarchy (more aggressive sharing, ≈33M unique params)* | | | | | | | | |
| Unshared, 33M | (Standard) | 3.627 | .437 | .292 | .604 | — | — | — |
| Unshared, 33M | (Wide) | 3.631 | .436 | .292 | .618 | — | — | — |
| Unshared, 33M | (Deep) | 3.719 | .423 | .291 | .607 | — | — | — |
| Hierarchy | (Standard) | 3.566 | .448 | .293 | .620 | +.011 | +.001 | +.016 |
| Hierarchy | (Wide) | 3.567 | .450 | .295 | .624 | +.014 | +.003 | +.006 |
| Hierarchy | (Deep) | 3.642 | .446 | .292 | .615 | +.023 | +.001 | +.008 |
| *250M scale* | | | | | | | | |
| Unshared (Full, 250M) | | **3.159** | **.559** | **.370** | **.655** | — | — | — |
| Unshared, 125M | | 3.296 | .510 | .335 | .647 | — | — | — |
| Unshared, 83M | | 3.389 | .492 | .319 | .633 | — | — | — |
| Sequence | | 3.262 | .532 | .347 | .646 | +.022 | +.012 | −.002 |
| Cycle | | — | .539 | .344 | .652 | +.029 | +.009 | +.005 |
| Cycle-Rev | | — | .521 | .345 | .657 | +.011 | +.010 | +.010 |
| Bulge | | — | .537 | .349 | .645 | +.028 | +.014 | −.003 |
| Hierarchy | | — | .489 | .331 | .629 | −.003 | +.012 | −.004 |
| *500M scale* | | | | | | | | |
| Unshared (Full, 500M) | | **3.030** | .611 | .416 | .675 | — | — | — |
| Unshared, 218M | | 3.184 | .546 | .362 | .662 | — | — | — |
| Unshared, 156M | | 3.268 | .525 | .343 | .648 | — | — | — |
| Sequence | | 3.128 | **.593** | **.389** | .668 | +.047 | +.027 | +.006 |
| Cycle | | 3.128 | .579 | .377 | **.671** | +.033 | +.015 | +.009 |
| Cycle-Rev | | 3.138 | .577 | .378 | .668 | +.031 | +.017 | +.007 |
| Bulge | | 3.143 | .561 | .376 | .659 | +.014 | +.014 | −.003 |
| Hierarchy | | 3.203 | .547 | .361 | .657 | +.022 | +.018 | +.009 |
| *1B scale* | | | | | | | | |
| Unshared (Full, 1B) | | **2.934** | **.655** | **.475** | **.705** | — | — | — |
| Unshared, 401M | | 3.068 | .613 | .415 | .678 | — | — | — |
| Unshared, 300M | | 3.117 | .578 | .394 | .666 | — | — | — |
| Sequence | | 3.005 | .630 | .440 | .682 | +.017 | +.025 | +.004 |
| Cycle | | 3.005 | .625 | .447 | .693 | +.012 | +.032 | +.015 |
| Cycle-Rev | | 3.009 | .639 | .437 | .691 | +.026 | +.022 | +.013 |
| Bulge | | 3.035 | .623 | .428 | .692 | +.010 | +.013 | +.014 |
| Hierarchy | | 3.082 | .598 | .411 | .682 | −.015 | −.004 | +.004 |

Table 3: **Deep sharing results across all model sizes.** Depth multipliers indicate how many times deeper the shared model is relative to the unshared baseline. "Skinny" models reduce width and increase depth further. All configurations are parameter-matched. **Bold** marks the best result per metric in each size group.

| Variant | Depth | Layers | Dim | LoRA $r$ | Val NLL | $\Delta$ NLL | ARC | $\Delta$ ARC | Hella | $\Delta$ Hella | PIQA | $\Delta$ PIQA |
|---|---|---|---|---|---|---|---|---|---|---|---|---|
| *100M scale* | | | | | | | | | | | | |
| Unshared (1×) | 1× | 12 | 768 | — | 3.435 | — | .485 | — | .309 | — | .627 | — |
| Shared (1.5×) | 1.5× | 18 | 768 | 174 | 3.447 | +.012 | .476 | −.009 | .308 | −.001 | .621 | −.006 |
| Very Deep (3×) | 3× | 36 | 768 | 87 | 3.418 | −.017 | **.505** | +.020 | .317 | +.008 | .616 | −.011 |
| Skinny (3.3×) | 3.3× | 40 | 512 | 207 | **3.384** | −.051 | .503 | +.018 | **.322** | +.013 | **.628** | +.001 |
| *250M scale* | | | | | | | | | | | | |
| Unshared (1×) | 1× | 16 | 1024 | — | 3.290 | — | .540 | — | .339 | — | .640 | — |
| Shared (1.5×) | 1.5× | 24 | 1024 | 233 | 3.286 | −.004 | .524 | −.016 | .340 | +.001 | .634 | −.006 |
| Very Deep (3×) | 3× | 48 | 1024 | 116 | 3.267 | −.023 | .543 | +.003 | .346 | +.007 | **.652** | +.012 |
| Skinny (3.75×) | 3.75× | 60 | 672 | 144 | **3.240** | −.050 | **.556** | +.016 | **.354** | +.015 | .642 | +.002 |
| *500M scale* | | | | | | | | | | | | |
| Unshared (1×) | 1× | 20 | 1280 | — | 3.172 | — | .559 | — | .368 | — | .653 | — |
| Shared (1.5×) | 1.5× | 30 | 1280 | 291 | 3.180 | +.008 | **.568** | +.009 | .369 | +.001 | .654 | +.001 |
| Very Deep (3×) | 3× | 60 | 1280 | 145 | 3.150 | −.022 | .563 | +.004 | **.376** | +.008 | **.663** | +.010 |
| Skinny (3×) | 3× | 60 | 896 | 193 | **3.146** | −.026 | .589 | +.030 | .383 | +.015 | .658 | +.005 |
| *1B scale* | | | | | | | | | | | | |
| Unshared (1×) | 1× | 28 | 1536 | — | — | — | **.617** | — | **.419** | — | **.684** | — |
| Shared (1.5×) | 1.5× | 42 | 1536 | 349 | **3.082** | — | .596 | −.021 | .410 | −.009 | .665 | −.019 |

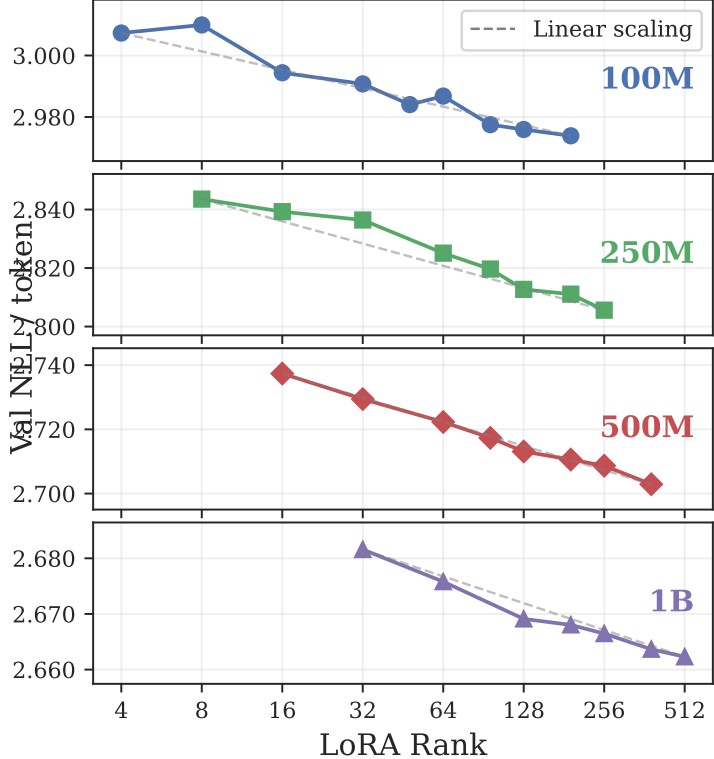

Figure 6: **Attention sharing LoRA rank vs. validation loss** (unmatched). Higher rank monotonically improves val loss, with clear diminishing returns. Each panel shows a single model size with a tight y-axis to reveal within-size trends. Dashed gray lines show linear scaling for reference.

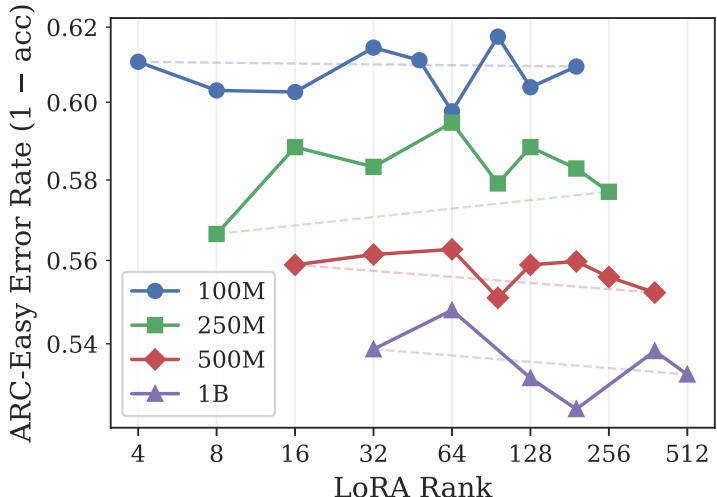

Figure 7: **Attention sharing LoRA rank vs. downstream error** (unmatched). ARC-Easy error rate (1 − accuracy; lower is better) is essentially flat across all ranks and sizes, indicating that attention sharing rank has negligible effect on downstream task performance.

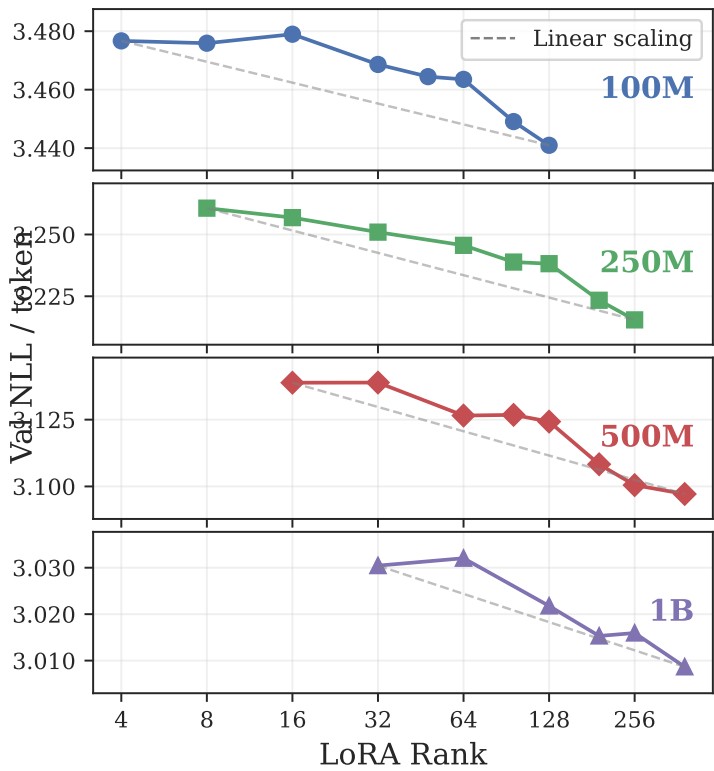

Figure 8: **Layer sharing LoRA rank vs. validation loss** (unmatched, bulge strategy). Similar diminishing-returns pattern to attention sharing. Each panel shows a single model size.

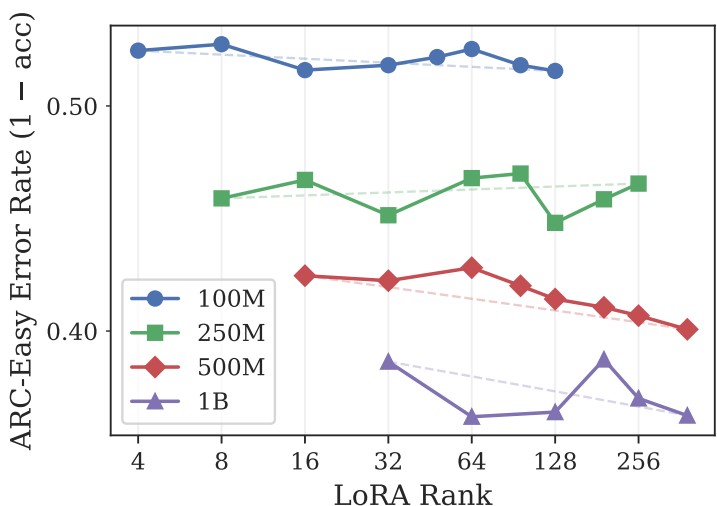

Figure 9: **Layer sharing LoRA rank vs. downstream error** (unmatched, bulge strategy). Unlike attention sharing, layer sharing shows meaningful downstream improvement with rank at 500M and 1B.

