# OpenReview forum: "Leveraging Low-Rank Structure for Effective Weight-Sharing in Language Models"
_ICLR.cc/2026/Workshop/Sci4DL — Sci4DL 2026_

### Official Review · Reviewer_R6Mn · 2026-02-27

**Fit:** 2
**Significance:** 2
**Confidence:** 2

**Summary:**

The submission investigates the strategy of weight-sharing to reduce the number of parameters in a large language model while not hurting its performance too much, as an alternative to scaling down the model's architecture. To mitigate the limitations of weight-sharing, the authors propose to encode additional weight differences as low-rank updates. They find that tying attention weights, as well as tying layers according to a variety of patterns, each supplemented with low-rank updates, reduces the number of parameters of the model while matching and sometimes improving the performance of parameter-matched unshared baselines.

**Strengths:**

The submission tackles the important problem of building parameter-efficient language models. The authors have conducted extensive experiments to identify the best weight-tying strategy, and benchmark their method against unshared baselines.

**Suggestions:**

Comments on the writing / figures / typos:
- On lines 88 and 92, reference is made to "Section 3(a/b)"; I think it should be "Figure 1(a/b)"
- In lines 87 to 95, does the sentence "The lowest-rank pairs are differences between attention head weights" mean that weights are more similar *within an attention layer* than *across layers*? If so, how is the comparison made? (It seems to me that plots 1(a) and 1(b) cannot be directly compared, as in one case it is the norm of the weights and in the other, the norm of the output).
- About Figure 1: 1) What is the relative energy plotted on the y axis? Does each curve of the left plot (e.g., l5_h1 - l5_h2) take into account the queries, the keys *and* the values of layer 5? I think that adding more information to the caption of the figure would increase its readability. 2) Also, it would be informative to visualize the 95% variance threshold on the plots (e.g., for the mean curve). 3) Finally, why is the mean so much higher than the other curves? It seems that the selected curves are not representative.
- In Figure 1(b), does a "layer" include both attention and perceptron?
- Line 194, how is the decomposition $EP$ of the embedding performed? (such a decomposition is not unique)

Other questions and comments:
- Queries, keys and values in multi-head attention are low-rank by construction (rank = input_dim / n_heads). Does the fact that differences between attention head weights are low-rank go beyond this observation, *i.e.*, is the rank of the difference smaller than the expected rank? I could not answer this question from Figure 1.
- About Figure 3, it seems that going below 27% for the embedding proportion does not hurt too much the loss, especially for larger models. For example, 18% seems well-suited for the 1B model. Underlining this would deliver a more precise message, in my opinion.
- The remark that Bulge performs better than Hierarchy is interesting. I think some references could be mentioned by the authors about it, as it has been previously observed that boundary layers have a different role than middle layers (see for example Persistent Topological Features in Large Language Models, Gardinazzi et al., 2025).

---

### Official Review · Reviewer_H87N · 2026-02-27

**Fit:** 3
**Significance:** 2
**Confidence:** 3

**Summary:**

This paper studies low-rank structure in a pretrained model, including (i) differences in attention-head weights within the same layer and (ii) differences in layer outputs across layers. Motivated by these observations, it proposes a weight-sharing strategy that shares weights across attention heads and layers via low-rank adapters. The experiement results show weight sharing matches or beats baselines at equal parameter count.  The paper also explores factorized embeddings and identifies an effective embedding rank through sweeping.

**Strengths:**

The analysis of low-rank structure in attention-head and layer-output differences is insightful, and it provides a clear motivation for the proposed method of weight sharing with low-rank adapters.

**Suggestions:**

1. The current experiments compare methods at the same parameter count. However, the proposed weight sharing method may require more FLOPs during training and inference, since weight sharing reduces parameters but not computation. It would be helpful to include comparisons under a fixed FLOPs budget. Alternatively, it may be helpful to clarify the primary resource constraint considered in this work, such as compute or GPU memory, and verify that weight sharing leads to end-to-end savings in that resource.

2. The paper analyzes low-rank structure in layer outputs. Since the proposed sharing operates on parameters, it would be helpful to validate low-rank structure directly in weight differences, or to justify why low-rank output differences support weight sharing at the parameter level.

3. Typo: l092, Section 3(b)  ->  Figure 1(b)?

---

### Official Review · Reviewer_G8Yw · 2026-02-27

**Fit:** 2
**Significance:** 2
**Confidence:** 2

**Summary:**

This paper proposes a parameter-efficient framework for small language models by sharing weights and modeling the differences with LoRA modules. It provides empirical justification to verify that differences between weight matrices in language models can be approximated by low-rank updates. Based on this observation, the authors train Llama-3-based models using attention-head and layer-wise sharing with LoRA adaptations and show that their performance can match that of an unshared model, but with fewer parameters. In addition, they investigate factorized embeddings by sweeping the embedding rank and report that an embedding proportion of 27% is nearly optimal in various settings.

**Strengths:**

1. The proposed method using weight tying and LoRA adaptations is well supported by empirical evidence.
2. It gives practical guidance for training parameter-efficient language models.

**Suggestions:**

1. **Experimental details**: In Table 1, models with parameter sizes from 100M to 1B are all trained with 10B tokens, which is not standard in the literature. According to the Chinchilla scaling laws, a 1B model is actually undertrained with 10B tokens. Training with an appropriate number of tokens would strengthen the experimental results of this paper.

2. **More principled choice of parameter sharing**: Although Section 3 provides empirical evidence of low-rank differences across layers, it does not give any guidance for the weight-sharing design in Section 4. For instance, it would be interesting to choose the rank in the LoRA module based on the SVD results or other diagnostics.

3. **Sensitivity analysis of rank in the LoRA module**: Most results compare two extreme choices of rank in the LoRA module. For example, in the upper part of Figure (b), they only report the Rank-22 (low rank) and Rank-144 (closer to full rank) cases. Since the choice of rank in LoRA is significant, it would be informative to investigate the effect of rank more systematically.

4. **Discussion of previous works**: The main ideas of the methodology including weight tying and LoRA modules have already been suggested in previous works. It would be helpful to compare this approach with other parameter-efficient techniques and conduct ablation studies. In particular, the recent work of [1] proposes similar weight sharing and LoRA techniques, so it should also be discussed.

[1] O'Neill et al., Low-Rank Key Value Attention, preprint.

---

### Decision · Program_Chairs · 2026-03-02

Accept